EMBO
Molecular Medicine

# miR-132 loss de-represses ITPKB and aggravates amyloid and TAU pathology in Alzheimer's brain

Evgenia Salta[1,2], Annerieke Sierksma[1,2], Elke Vanden Eynden[1,2] & Bart De Strooper[1,2,3,*]

## Abstract

microRNA-132 (miR-132) is involved in prosurvival, anti-inflammatory and memory-promoting functions in the nervous system and has been found consistently downregulated in Alzheimer's disease (AD). Whether and how miR-132 deficiency impacts AD pathology remains, however, unaddressed. We show here that miR-132 loss exacerbates both amyloid and TAU pathology via inositol 1,4,5-trisphosphate 3-kinase B (ITPKB) upregulation in an AD mouse model. This leads to increased ERK1/2 and BACE1 activity and elevated TAU phosphorylation. We confirm downregulation of miR-132 and upregulation of ITPKB in three distinct human AD patient cohorts, indicating the pathological relevance of this pathway in AD.

**Keywords** Alzheimer's; amyloid; ITPKB; microRNA-132; TAU

**Subject Category** Neuroscience

## Introduction

The pathogenic process in Alzheimer's disease (AD) involves a long cellular phase during which intricate feedback and feed-forward cascades between distinct cell types affect the homeostasis of the brain. This progressively leads to the clinical stage of the disease (De Strooper & Karran, 2016). MicroRNAs (miRNAs) keep the expression of various genes in check and are typically part of molecular homeostasis mechanisms. While the expression of miRNAs is disturbed in AD brain (Hébert *et al*, 2008; Lau *et al*, 2013), it remains largely unknown how these aberrations are induced and whether they play a role in disease progression (Salta & De Strooper, 2012). Even if the disruption of miRNA expression is part of the general pathogenic process induced by accumulating toxic cues, it remains important to investigate whether loss or gain of function of particular miRNAs has also a specific functional impact on AD pathology. miRNAs that are systematically and early downregulated in the disease course are of particular interest in that regard. MicroRNA-132 (miR-132) is one of the few miRNAs that are consistently and robustly downregulated in AD brain (Cogswell *et al*, 2008; Hebert *et al*, 2013; Lau *et al*, 2013;

Wong *et al*, 2013; Smith *et al*, 2015), suggesting a functional involvement in the pathogenic process. miR-132 has neuroprotective properties regulating the prosurvival PI3K-AKT pathway, which represses a death signaling cascade that involves FOXO3a, P300, and PTEN in primary hippocampal and cortical neurons (Wong *et al*, 2013). Moreover, miR-132 expression levels are negatively correlated to hyperphosphorylated TAU aggregates in the cortex of AD patients (Lau *et al*, 2013) and miR-132 knockout increases TAU phosphorylation and aggregation in a triple transgenic AD mouse model (Smith *et al*, 2015). Notably, miR-132 deficiency in AD brain might have additional neurotoxic effects as miR-132 is involved in neuronal plasticity and synaptic function (Edbauer *et al*, 2010; Bicker *et al*, 2014; Salta *et al*, 2014), it has been implicated in neuroinflammation and the regulation of acetylcholinesterase expression (Shaked *et al*, 2009), while activity-induced CREB-dependent miR-132 transcription also contributes to memory formation and cognition (Hansen *et al*, 2013). Overall these observations suggest that loss of miR-132 could play a pivotal role in several aspects of AD. However, to date, little hard data are available providing factual support to the hypothesis that miR-132 is part of the disease process.

In this study, we set out to characterize the functional impact of miR-132 deficiency on AD pathology in brain and to begin to dissect the molecular networks underlying these effects. We demonstrate here that downregulation of miR-132 aggravates both amyloid and TAU pathology in AD mice and that it regulates the expression of inositol 1,4,5-trisphosphate 3-kinase B (ITPKB), a regulator of BACE1 activity and TAU phosphorylation (Stygelbout *et al*, 2014). We suggest that loss of miR-132 is part of a feed forward loop enhancing the biochemical stress that drives the disease.

## Results

### *In vivo* manipulation of miR-132 levels in the hippocampus of APPPS1 mice

We previously observed miR-132 downregulation at Braak stage III in AD brain (Lau *et al*, 2013), suggesting a relatively early role of this miRNA in the disease process. To asses this further, we downregulated miR-132 using intracerebroventricular (ICV) injections of locked nucleic acid (LNA)-modified, 3' cholesterol-conjugated antisense

1   VIB Center for the Biology of Disease, Leuven, Belgium
2   Center for Human Genetics, Universitaire ziekenhuizen and LIND, KU Leuven, Belgium
3   Institute of Neurology, University College London, London, UK
    *Corresponding author. Tel: +32 16373246; E-mail: bart.destrooper@cme.vib-kuleuven.be

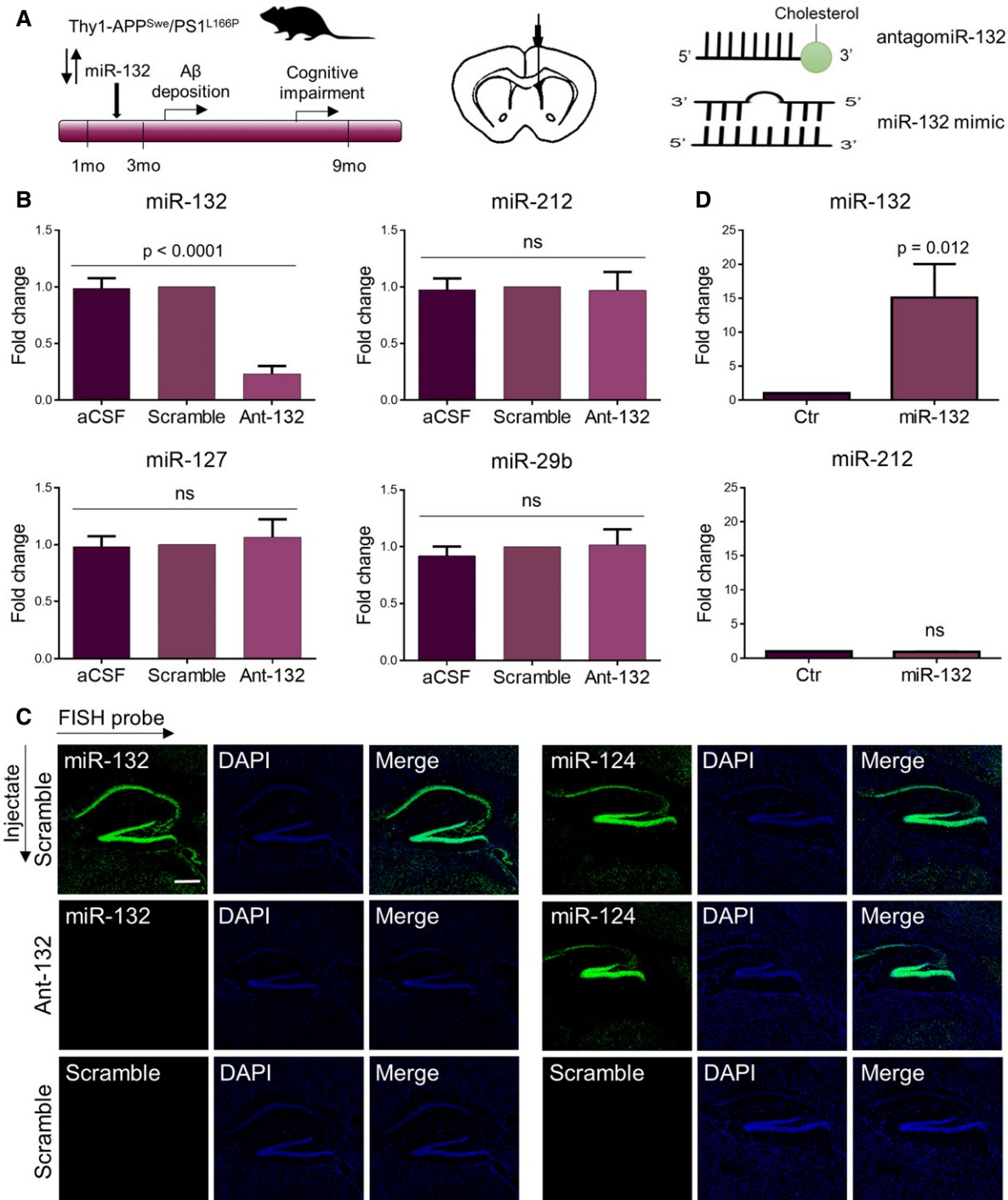

**Figure 1. Efficiency and specificity of *in vivo* down- and upregulation of miR-132.**

A  Experimental scheme of antagomiR-132 and miR-132 mimic injections into the lateral ventricle of 2-month-old APPPS1 mice.

B  Semi-quantitative PCR of miR-132 and negative control miRNAs in the hippocampus of antagomiR-132-injected mice (ant-132) in comparison with control-injected animals (aCSF and scramble) at 6 months of age. Sample size, *n* = 9 per group.

C  FISH of miR-132 and negative control miR-124 in the hippocampus of ant-132- and scramble-injected mice. Scramble probes were used as FISH negative controls. Scale bar, 500 μm.

D  Semi-quantitative PCR of miR-132 and miR-212 in the hippocampus of 3-month old miR-132 mimic-injected mice (miR-132) compared to animals injected with a negative control oligonucleotide (Ctr). Sample size, *n* = 6 per group.

Data information: Values were normalized to scramble- (B) or control-injected group (D) and presented as mean ± SEM. In (B), one-way ANOVA was used, while in (D), Student's *t*-test was applied.

   

antagomiR-132 oligonucleotides in 2-month-old APPPS1 mice. These mice co-express human-mutated APP$^{Swe}$ (KM670/671NL APP) and human-mutated presenilin 1 (L166P) (Fig 1A). This widely used AD model [Tg(Thy1-APPSw, Thy1-PSEN1*L166P) 21Jckr] typically shows amyloid deposition in the hippocampus at 3–4 months and cognitive impairment at 7–8 months of age (Radde *et al*, 2006; Serneels *et al*, 2009). Monthly ICV injections of antagomiR-132 resulted in an approximately 80% decrease of miR-132 levels at 6 months of age (Fig 1B and C). A miR-132 scramble oligonucleotide did not have any effect on miR-132 expression, similarly to injections with only artificial cerebrospinal fluid (aCSF) used as carrier. No changes in the expression of miR-212—a miRNA transcribed in the same cluster as miR-132—or of three other, unrelated miRNAs, that is miR-127, miR-29b (Fig 1B), and miR-124 (Fig 1C), were observed.

We additionally performed overexpression experiments using double-stranded miR-132 mimic oligonucleotides. Four ICV injections of 150 pmol miR-132 mimic with one-week interval led to a 15-fold upregulation of miR-132 in hippocampus at 3 months of age, again without affecting the levels of the closely related miR-212 (Fig 1D).

### miR-132 regulates soluble and insoluble Aβ

We assessed the levels of Aβ after up- or downregulation of miR-132. Amyloid immunostaining in the ant-132-injected animals revealed a twofold and a 1.7-fold increase of plaque burden in hippocampus and cortex, respectively (Fig 2A and B). We next used ELISA to measure hippocampal Aβ$_{40}$ and Aβ$_{42}$ levels. Both peptides were significantly increased in the cytoplasmic and extracellular TBS-soluble and the

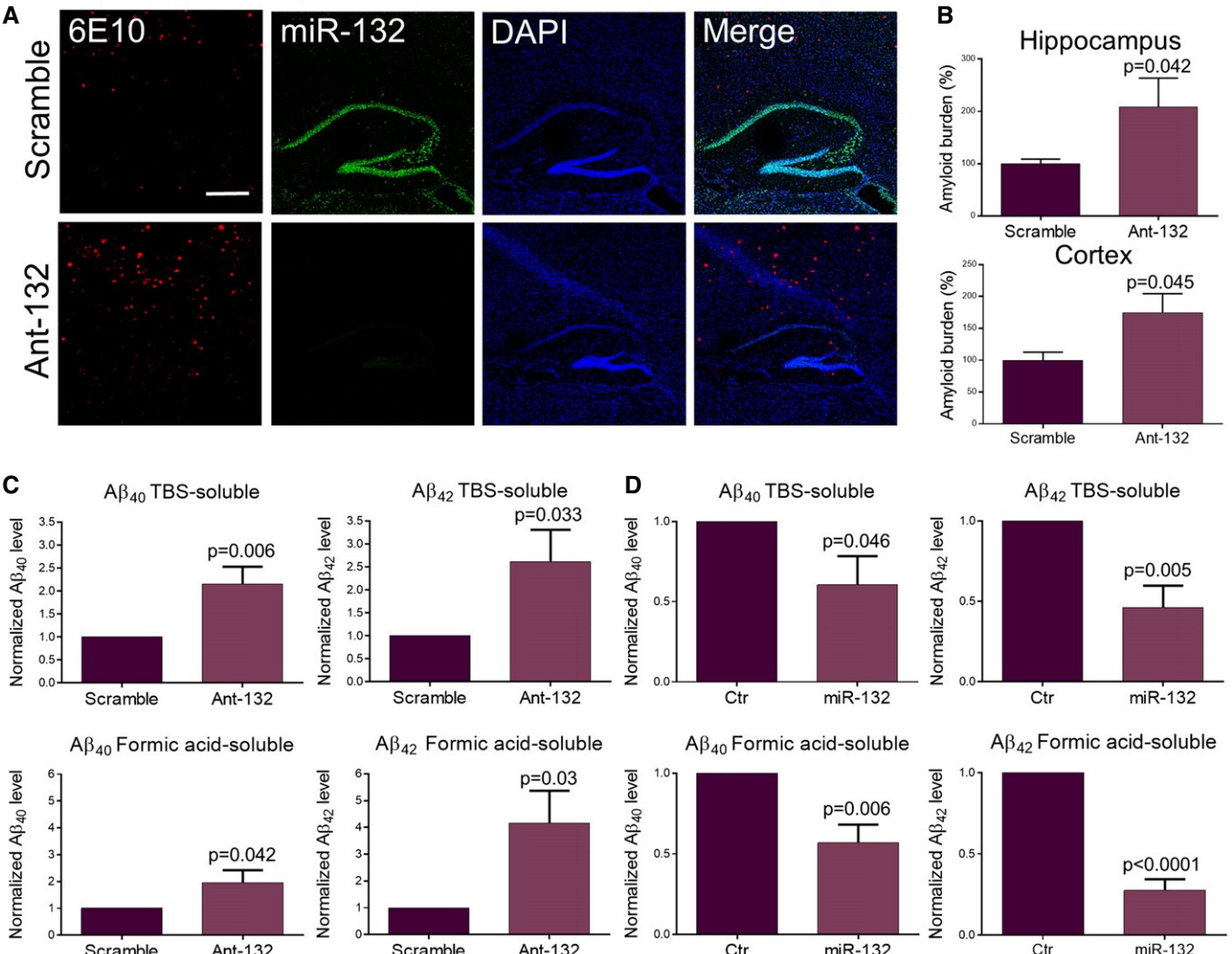

**Figure 2.  Effect of miR-132 on Aβ accumulation.**

A    Amyloid staining (6E10) combined with miR-132 FISH in ant-132- and scramble-injected mice at 6 months of age. Scale bar, 500 μm.

B    Quantification of amyloid burden in hippocampus and cortex of ant-132-injected and control animals (see Materials and Methods). Sample size, *n* = 4 per group.

C, D   ELISA of soluble (TBS) and insoluble (Formic acid) Aβ$_{40}$ and Aβ$_{42}$ levels in the hippocampus of ant-132- (C) and miR-132-injected (D) animals at 6 and 3 months of age, respectively. Sample size, *n* = 6 per group.

Data information: In (B–D) values were normalized to scramble- (B, C) or control-injected group (D) and presented as mean ± SEM. Student's *t*-test was used.

   

formic acid-soluble fractions in response to miR-132 downregulation (Fig 2C). In contrast, a significant decrease of soluble and insoluble $A\beta_{40}$ and $A\beta_{42}$ was observed upon miR-132 ectopic expression in hippocampus (Fig 2D). These data indicate an inhibitory effect of miR-132 over soluble and aggregated Aβ species formation.

## miR-132 controls TAU phosphorylation

Among the most prominent kinases involved in TAU hyperphosphorylation in AD are ERK1/2, CDK5, and GSK3B (Mandelkow *et al*, 1995), and *in silico* analysis (Targetscan 7.0) predicts ERK1, ERK2, GSK3b, and TAU itself as putative miR-132 targets in human and mouse. Total TAU levels were, however, not affected following miR-132 down- or upregulation (Fig 3A and B top panel) contrary to a previous report (Smith *et al*, 2015). TAU phosphorylation levels were, in contrast, significantly altered in antagomiR-132- or miR-132 mimic-injected mice as assessed by Western blot using antibody AT8 against phosphorylated Ser-202/Thr-205 and antibody AT270 against phosphorylated Thr-181 (Mandelkow & Mandelkow, 2012) (Fig 3A and B middle and bottom panels). These two epitopes can be phosphorylated by ERK1/2, CDK5, and GSK3b (Fig EV1). However, no change in the expression levels of these kinases was observed upon miR-132 downregulation in APPPS1 hippocampus (Fig EV2). We further assessed the phosphorylation of two additional sites in TAU, which are phosphorylated by CDK5 and GSK3B but not by ERK1/2, namely Thr-212/

Ser-214 (recognized by antibody AT100) and Thr-231 (recognized by antibody AT180) (Mandelkow & Mandelkow, 2012; Fig EV1). Interestingly, phosphorylation levels of these epitopes were unaffected by miR-132 ectopic expression, which, together with the positive AT8/ AT270 staining, suggested that ERK1/2 might be part of the kinases involved in the miR-132-mediated effect on TAU phosphorylation.

## miR-132 target identification

While the previous experiments demonstrate that miR-132 downregulation increases Aβ generation and TAU phosphorylation, the targets mediating these effects remain unknown. In a first approach to make a choice among the 1,332 predicted miR-132 targets conserved among human and mouse (Targetscan 7.0), we made use of data previously obtained in six healthy (Braak stage 0-I) and six diseased human AD (Braak stage V-VI) prefrontal cortex (PFC) samples. We quantified the levels of miR-132 in the different brains using the next generation RNA sequencing dataset published in Lau *et al* (2013) and the levels of the different candidate mRNAs as deduced from the microarray transcriptome study performed on the same samples (Bossers *et al*, 2010) and correlated miR-132 expression and predicted miR-132 target levels (Fig 4A) (see Materials and Methods). This approach yielded a list of 19 predicted miR-132 targets whose expression is significantly upregulated and anticorrelated with miR-132 levels in AD (Table 1). We then assessed with real-time PCR

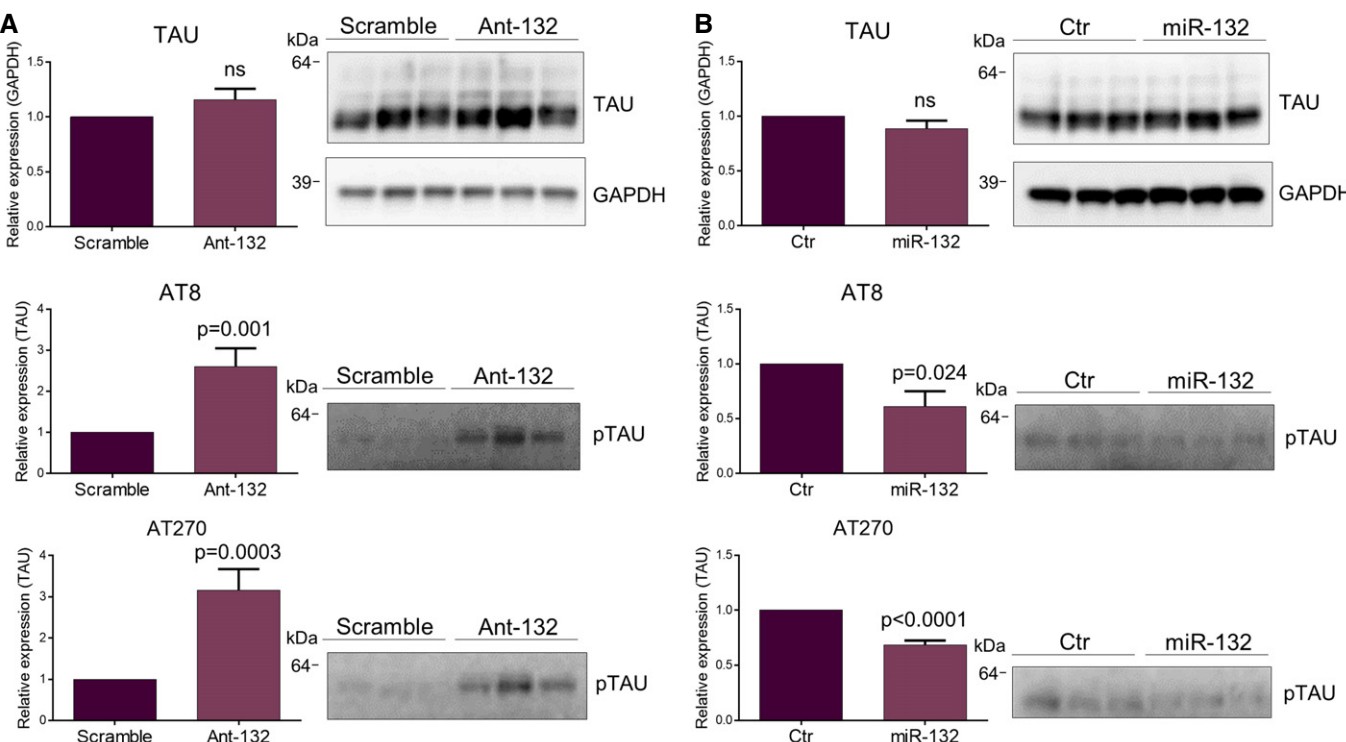

**Figure 3.   Regulatory effect of miR-132 on TAU expression and phosphorylation.**

A, B    Western blot analysis of total TAU and pTAU (AT8, AT270) levels upon miR-132 knockdown (A) or overexpression (B) at 6 and 3 months of age, respectively. Sample size, *n* = 9 per group (A) and *n* = 6 per group (B). Values were normalized to the respective control groups and presented as mean ± SEM. Student's *t*-test was used.

Source data are available online for this figure.

    

which of these transcripts additionally became upregulated in the APPPS1 mice upon miR-132 downregulation (Fig 4B). We set as threshold a 1.3-fold change above baseline levels (Selbach *et al*, 2008) and found five mRNAs that responded to that criterion: *Taf4*, *Arid1A*, *Erbb2ip*, *ItpkB*, and *Kccn3*. A search in PubMed using as keywords each transcript's name in combination with *Alzheimer*, *amyloid*, *Abeta*, or *TAU* yielded no results for *Taf4*, *Arid1A*, *Erbb2ip*, and *Kcnn3*. However, strikingly, one of these candidates, the inositol 1,4,5-trisphosphate 3-kinase B (ITPKB), was recently shown to induce both Aβ aggregation and TAU phosphorylation in 5XFAD mice (APP K670N/M671L (Swedish) + I716V (Florida) + V717I (London), and PS1 M146L + L286V) (Stygelbout *et al*, 2014). Since such a target would theoretically explain the observed impact of miR-132 deficiency on Aβ and pTAU in APPPS1 mice, we set out to further explore the possible regulatory interplay between miR-132 and ITPKB.

### The miR-132 target ITPKB mediates miR-132 effects on Aβ

The 3′UTR of ITPKB contains one predicted miR-132 binding site conserved among human and mouse and an additional one which is unique to human ITPKB. We used a luciferase reporter construct containing the human ITPKB 3′UTR to co-transfect HEK293 cells along with a miR-132 mimic or a negative control oligonucleotide. Co-transfection with the miR-132 mimic led to an approximate luminescence repression of 70% compared to the negative control-transfected cells (Fig 5A). Mutating the two predicted miR-132 binding sites in the 3′UTR of ITPKB completely abolished the effect on luciferase enzymatic activity (Fig 5A), indicating that ITPKB is a direct miR-132 target *in vitro*. Since ITPKB has been shown to affect Aβ generation (Stygelbout *et al*, 2014), we assessed whether ITPKB is necessary and/or sufficient to induce the effect of miR-132 on Aβ levels. Transfection of HEK293 cells overexpressing human APP$^{Swe}$ with a miR-132 antisense oligonucleotide led to a significant increase of both Aβ$_{40}$ and Aβ$_{42}$ confirming our *in vivo* findings (Fig 5B). Interestingly, downregulation of ITPKB using an siRNA oligonucleotide resulted in a reduction of Aβ levels pointing toward the pro-amyloidogenic role of ITPKB (Fig 5B). Moreover, simultaneous downregulation of ITPKB rescued the miR-132 knock down effect on Aβ levels (Fig 5B). Knockdown efficiency of miR-132 and ITPKB is shown in Fig EV3. We then assessed whether altering ITPKB levels was sufficient to exert an effect on Aβ levels *in vivo*. Indeed, downregulation of ITPKB using ICV injections of an ITPKB siRNA resulted in significantly decreased insoluble Aβ levels in APPPS1 hippocampus (Fig 5C and D), a finding that mimics the effects observed upon miR-132 overexpression and indicates the possible functional involvement of ITPKB in the miR-132-dependent effect on Aβ. Finally, to dissect the *in vivo* regulatory relationship between miR-132 and ITPKB, we assessed ITPKB expression in the hippocampus of antagomiR-132- and miR-132 mimic-injected APPPS1 mice. ITPKB was significantly upregulated following miR-132 downregulation (Fig 5E), while miR-132 overexpression significantly repressed ITPKB levels (Fig 5F) indicating that, also *in vivo*, ITPKB is under the regulatory control of miR-132.

### miR-132 regulates ERK1/2 and BACE1 enzymatic activity

It has been previously shown that ITPKB has a dual regulatory role in relation to AD pathology: It can affect BACE1 activity leading to more Aβ generation, while at the same time, it promotes TAU

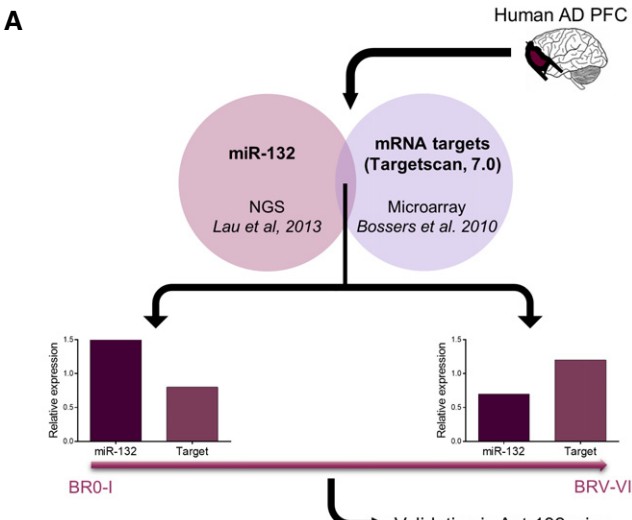

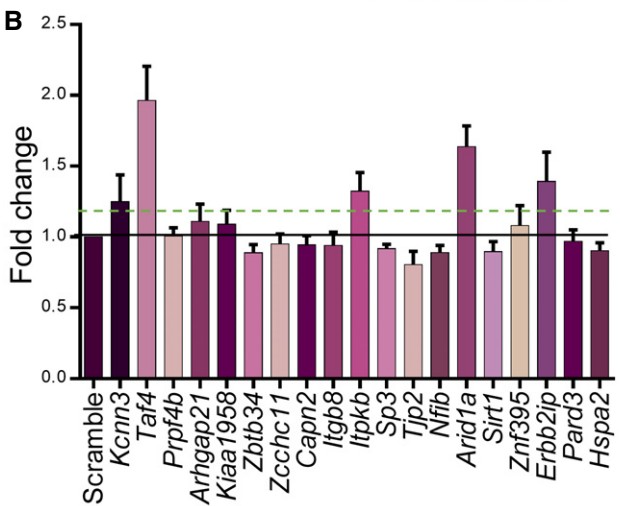

**Figure 4.  miR-132 target screening and validation.**

A   Correlation analysis process: miRNA and mRNA profiling datasets were cross-checked for miR-132 predicted targets that are significantly anticorrelated to miR-132 expression and significantly upregulated at late stages of AD.

B   Positive hits were validated by semi-quantitative qPCR in ant-132-injected mouse hippocampus. Black line indicates mean of scramble control set to 1. Dashed line indicates chosen threshold of 1.3-fold. Sample size, *n* = 9 per group. Values were normalized to scramble-injected group and presented as mean ± SEM.

phosphorylation via phosphorylation and activation of ERK1/2 (Stygelbout *et al*, 2014). No differences between antagomiR-132- and control-injected mice were observed with regard to total APP and BACE1 levels (Fig EV4). Interestingly, APP CTFβ and soluble APPβ (sAPPβ), the two products derived from the proteolytic processing of APP by BACE1, were indeed elevated upon miR-132 downregulation in the hippocampus of the APPPS1 mice, while a significant decline was observed upon miR-132 upregulation (Fig 6A and B), indicating that miR-132 negatively regulates BACE1 activity. Moreover, increased levels of phosphorylated ERK1/2 were observed in the antagomiR-132-injected mice, while reversely, phosphorylation of ERK1/2 decreased in the miR-132 mimic-injected

Table 1.    Predicted miR-132 targets that were significantly upregulated and anticorrelated with miR-132 expression in human AD prefrontal cortex.

| Transcript ID | | Pearson's correlation | | | Differential expression | |
|---|---|---|---|---|---|---|
| Symbol | NCBI ID | r | P-value | BH adj P-value | P-value | BH adj P-value |
| KIAA1958 | AB075838 | −0.878 | 0.0002 | 0.0175 | 6.3E−05 | 0.0142 |
| ZBTB34 | AB082524 | −0.857 | 0.0004 | 0.0285 | 2.8E−09 | 0.0001 |
| ZCCHC11 | NM_001009881 | −0.846 | 0.0005 | 0.0294 | 9.0E−05 | 0.0165 |
| CAPN2 | NM_001748 | −0.768 | 0.0036 | 0.0443 | 3.7E−04 | 0.0277 |
| ITGB8 | NM_002214 | −0.765 | 0.0038 | 0.0443 | 7.5E−05 | 0.0156 |
| ITPKB | NM_002221 | −0.766 | 0.0036 | 0.0443 | 2.4E−04 | 0.0233 |
| KCNN3 | NM_002249 | −0.814 | 0.0013 | 0.0362 | 1.1E−04 | 0.0186 |
| SP3 | NM_003111 | −0.765 | 0.0037 | 0.0443 | 1.0E−03 | 0.0443 |
| TAF4 | NM_003185 | −0.884 | 0.0001 | 0.0163 | 3.8E−06 | 0.0035 |
| PRPF4B | NM_003913 | −0.783 | 0.0026 | 0.0421 | 2.0E−04 | 0.0220 |
| TJP2 | NM_004817 | −0.731 | 0.0069 | 0.0555 | 1.2E−03 | 0.0478 |
| NFIB | NM_005596 | −0.764 | 0.0038 | 0.0443 | 7.3E−04 | 0.0386 |
| ARID1A | NM_006015 | −0.859 | 0.0003 | 0.0285 | 1.1E−04 | 0.0186 |
| SIRT1 | NM_012238 | −0.908 | 0.0000 | 0.0112 | 5.4E−07 | 0.0013 |
| ZNF395 | NM_018660 | −0.792 | 0.0021 | 0.0403 | 4.4E−04 | 0.0302 |
| ERBB2IP | NM_018695 | −0.784 | 0.0026 | 0.0420 | 3.6E−04 | 0.0276 |
| PARD3 | NM_019619 | −0.802 | 0.0017 | 0.0370 | 1.1E−04 | 0.0182 |
| ARHGAP21 | NM_020824 | −0.820 | 0.0011 | 0.0354 | 1.2E−04 | 0.0190 |
| HSPA2 | NM_021979 | −0.886 | 0.0001 | 0.0161 | 2.1E−05 | 0.0076 |

P-values were adjusted using the Benjamini–Hochberg correction. Differential expression data were deduced from Bossers et al (2010).

APPPS1 hippocampus (Fig 6C), suggesting that miR-132 acts upstream of ITPKB in a cascade eventually regulating ERK1/2 phosphorylation.

### ITPKB colocalizes with amyloid plaques and neurofibrillary tangles in human AD brain and exhibits a mutually exclusive expression pattern to miR-132

We further validated miR-132 levels in an independent set of human AD prefrontal cortex samples. miR-132 was approximately twofold downregulated in AD compared to non-demented control samples in agreement with previous work (Lau et al, 2013; Fig EV5). miR-124 was previously reported not to change in human AD prefrontal cortex (Lau et al, 2013) and was therefore used as negative control. Interestingly, ITPKB colocalized with amyloid plaques (Fig 7A). We then assessed the in situ expression patterns of miR-132 and ITPKB in neurons bearing NFTs in AD prefrontal cortex (Fig 7B and C). Low miR-132 expression was observed in cells displaying ITPKB and pTAU accumulation, while miR-132 signal was high in cells with lower ITPKB+- and pTAU+-immunolabeling. This was not observed for miR-124 expression. ITPKB levels were elevated in cells with strong pTAU immunoreactivity.

### The miR-132/ITPKB pathway is disrupted in human AD brain

We finally assessed the expression of the different molecular players in 39 late-stage AD and 15 control hippocampal samples. These brains were part of the patient cohorts in which we originally

reported the downregulation of miR-132 (Lau et al, 2013). We confirmed miR-132 deficiency in these samples by real-time PCR (Fig 8). Interestingly, along elevated phosphorylated ERK1/2 and phosphorylated TAU levels, we also observed significantly increased ITPKB expression (Fig 8). Notably, ITPKB levels were significantly correlated to phosphorylated TAU (AT270) levels (Pearson's r = 0.7; P = 0.002). No changes in total ERK1/2 or TAU expression were observed.

## Discussion

We show here that loss of miR-132 in the context of an AD mouse model affects the two arms of the biochemical cascade that leads to the well-known AD pathology. miR-132 knockdown in APPPS1 mice increases soluble and insoluble Aβ and amyloid burden, while it also enhances TAU phosphorylation in hippocampus. These effects are reversed upon miR-132 overexpression, suggesting a direct relation between miR-132 levels and these biochemical phenotypes. Thus, the strong miR-132 downregulation previously observed in AD brain is not only a consequence of the disease process but also contributes to the biochemical alterations that characterize the pathology. The dual effect on Aβ and TAU is surprising and is apparently—partially—mediated by inositol 1,4,5-trisphosphate 3-kinase B or ITPKB. Interestingly, ITPKB mRNA and protein levels increase up to threefold in human AD frontal cortex (Bossers et al, 2010; Saetre et al, 2011; Stygelbout et al, 2014). Here, we confirm the significant ITPKB upregulation along with the strong miR-132

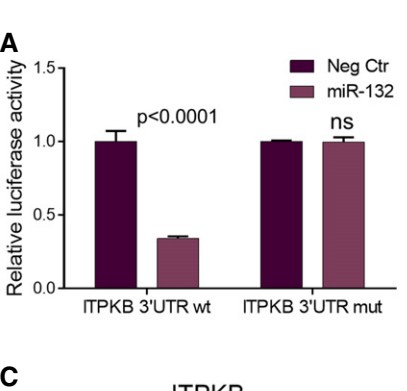

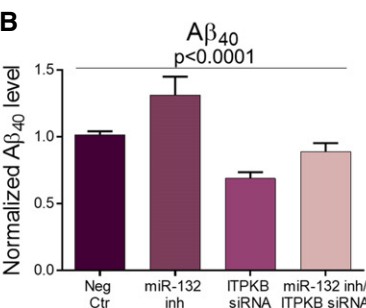

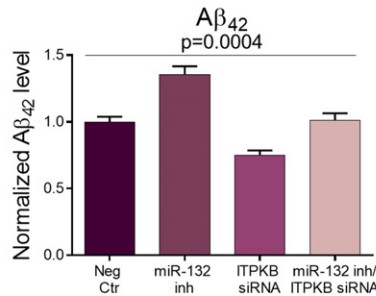

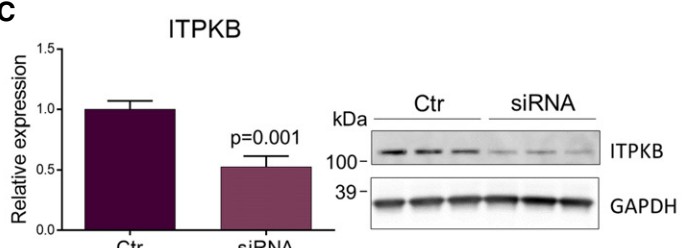

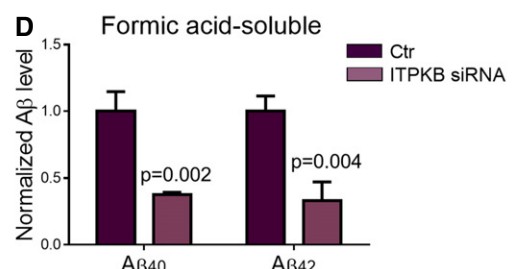

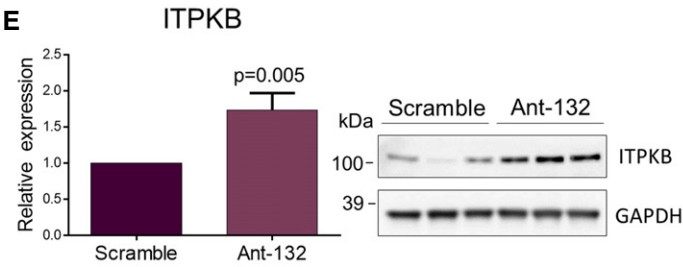

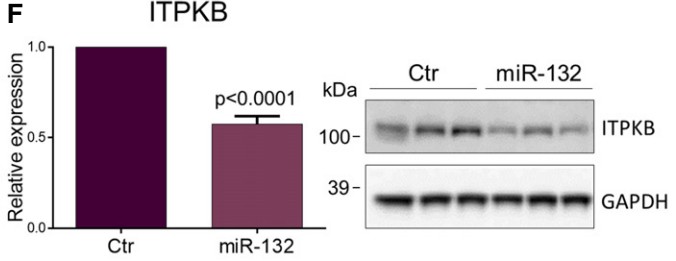

**Figure 5.  miR-132 regulation over ITPKB.**

A    Luciferase reporter assay of wild-type (wt) and mutant (mut) ITPKB 3′UTR in HEK293 cells co-transfected with a synthetic miR-132 (miR-132) or a negative control (Neg Ctr) oligonucleotide.

B    ELISA of $A\beta_{40}$ and $A\beta_{42}$ levels in HEK293-APP$^{swe}$ cells transfected with a miR-132 antisense oligonucleotide (miR-132 inh), an siRNA against ITPKB (ITPKB siRNA) or both.

C    Western blot analysis of ITPKB knockdown in APPPS1 hippocampus using an siRNA oligonucleotide against ITPKB. Sample size, $n$ = 6 per group. Values were normalized to control-injected group (Ctr) and presented as mean ± SEM.

D    ELISA of insoluble (formic acid soluble) $A\beta_{40}$ and $A\beta_{42}$ levels in the hippocampus of ITPKB siRNA- and control-injected animals at 3 months of age. Sample size, $n$ = 6 per group. Values were normalized to control group and presented as mean ± SEM.

E, F  Western blot analysis of ITPKB levels upon miR-132 down (E)- or upregulation (F) at six and three months of age, respectively. Sample size, $n$ = 9 per group for miR-132 downregulation and $n$ = 6 per group for miR-132 overexpression. Values were normalized to control groups and presented as mean ± SEM.

Data information: The assays in (A and B) were performed in three independent experiments, each in triplicates. In (A, C–F) Student's *t*-test was used, while in (B), one-way ANOVA was employed.

Source data are available online for this figure.

downregulation in AD brain. Moreover, up- or downregulation of miR-132 in mouse hippocampus led to a direct down- or upregulation of ITPKB, respectively. Intriguingly, this kinase is known to phosphorylate ERK1/2 (Wen *et al*, 2004; Maréchal *et al*, 2007) leading to increased TAU phosphorylation and at the same time enhanced BACE1 activity (and Aβ accumulation) in another mouse model of AD (Stygelbout *et al*, 2014). Along these lines, we report that ITPKB deficiency *in vivo* results in a significant decrease in Aβ levels in AD mouse hippocampus. The direct link between miR-132 and ITPKB was further confirmed in a genetic occlusion experiment in cell culture demonstrating that the effect of downregulation of miR-132 on Aβ generation is neutralized by downregulating ITPKB at the same time (Fig 5B).

We and others previously suggested that miR-132 directly affects TAU expression as *MAPT* mRNA contains a miR-132 binding site (Lau *et al*, 2013; Smith *et al*, 2015). We could not demonstrate such an effect here, although Smith *et al* (2015) recently reported elevated mouse and human TAU expression in both wild-type and triple transgenic (APP$^{Swe}$/PSEN$^{M146V}$/TAU$^{P301L}$, 3xTg-AD) mice following genetic deletion of the miR-132/212 cluster. Obviously, we induce here miR-132 knockdown in young adult mice, while the genetic knockout is already present at birth, which could explain this apparent discrepancy. It should be pointed out, however, that in the latter model, the human TAU transgene does not contain the *MAPT* 3′UTR, and therefore, it is not clear how the increased human TAU levels in the miR-132 knockout mice are

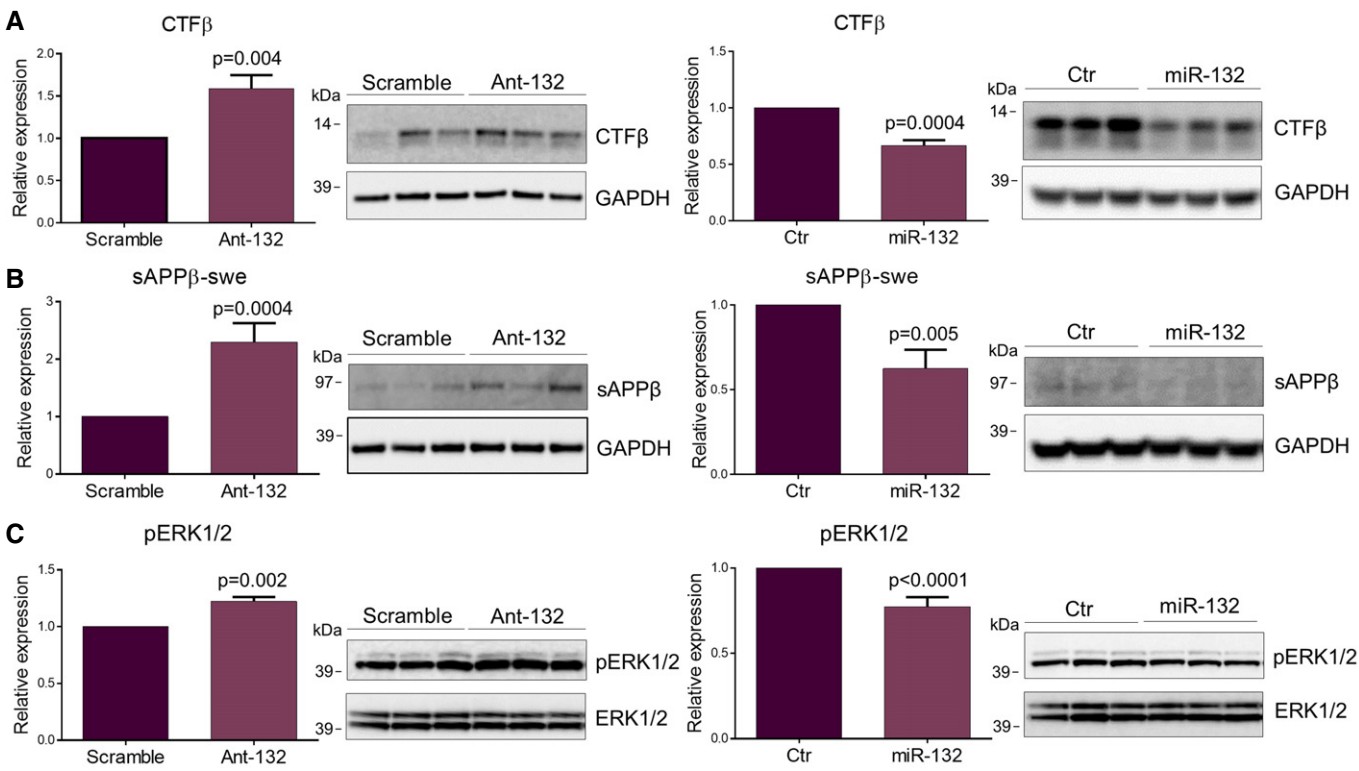

**Figure 6.   Effect of miR-132 regulation on BACE1 and ERK1/2 activity.**

A–C   Western blot analysis of CTFβ (A), sAPPβ-swe (B) and phosphorylated ERK1/2 (pERK1/2) (C) levels upon miR-132 down (ant-132) (left panel)- or upregulation (miR-132) (right panel) in APPPS1 hippocampus at 6 and 3 months of age, respectively. Sample size, n = 9 per group for miR-132 downregulation and n = 6 per group for miR-132 overexpression. Values were normalized to respective control groups and presented as mean ± SEM. Student's t-test was used.

Source data are available online for this figure.

explained. Our data indicate that the endogenous TAU is not directly affected by miR-132 regulation, but that, indirectly, increased activity of ERK1/2, induced by ITPKB, could explain the increased TAU phosphorylation.

The second AD-related effect of miR-132 downregulation, the increase in Aβ generation, is also—at least partially–mediated by increased ITPKB levels. This effect has the signature of increased BACE1 activity as APP CTFβ and sAPPβ were increased, as well. A previous report had already indicated that ITPKB activates BACE1 (Stygelbout *et al*, 2014). Interestingly, ERK1/2 has previously been shown to affect membrane lipid composition and thereby promote BACE1 enzymatic activity via phosphorylation of sphingosine kinase in neurons (Takasugi *et al*, 2011). Further supporting the possible relevance of these observations in human brain, we found increased levels of phosphorylated sphingosine kinase in the AD hippocampi compared to the controls (Fig 8).

The real novelty of our work stems from the insight that miR-132 is upstream of ITPKB, which links our previous observation of miR-132 downregulation in AD to a pathologically relevant mechanism. The link between miR-132 and ITPKB, ERK1/2 activity, APP processing, Aβ accumulation and TAU phosphorylation is strong as shown by the consistent effects upon *in vivo* up- and downregulation of miR-132 in the brain of an AD mouse model. We further

employed three independent sets of AD patient samples to explore to which extent the proposed miR-132/ITPKB pathway may also occur in AD brain. Notably, ITPKB significantly increased in late-stage AD hippocampal samples, in which miR-132 levels were found to be decreased by approximately 50%. Moreover, in the same set of samples, we were able to confirm enhanced levels of activated ERK1/2 (pERK1/2) and pTAU. ERK is required for TAU hyperphosphorylation in mice (Le Corre *et al*, 2006), while ERK1/2 activity, but not total protein levels, was previously found to be elevated in human AD brain (Ferrer *et al*, 2001) concomitantly with initial TAU deposition, reflecting one of the earliest events in disease pathogenesis (Stygelbout *et al*, 2014). This is remarkably congruent with the miR-132 downregulation at the early Braak stage III as previously reported (Lau *et al*, 2013).

We report here that the well-established miR-132 loss in AD brain has a presumably early, dual effect on key biochemical aspects of pathogenesis: It aggravates both amyloid and TAU pathology likely, and at least partially, via direct regulation of the kinase ITPKB. This mechanism amplifies both pathologies. Strategies to simultaneously tackle both amyloid and TAU pathways in AD would possibly represent a highly efficient approach to halt the pathogenic process. Therefore, and given the versatility and the increasing know-how in using miRNAs as therapeutics, it is attractive to speculate on the potential use of miR-132 mimics to

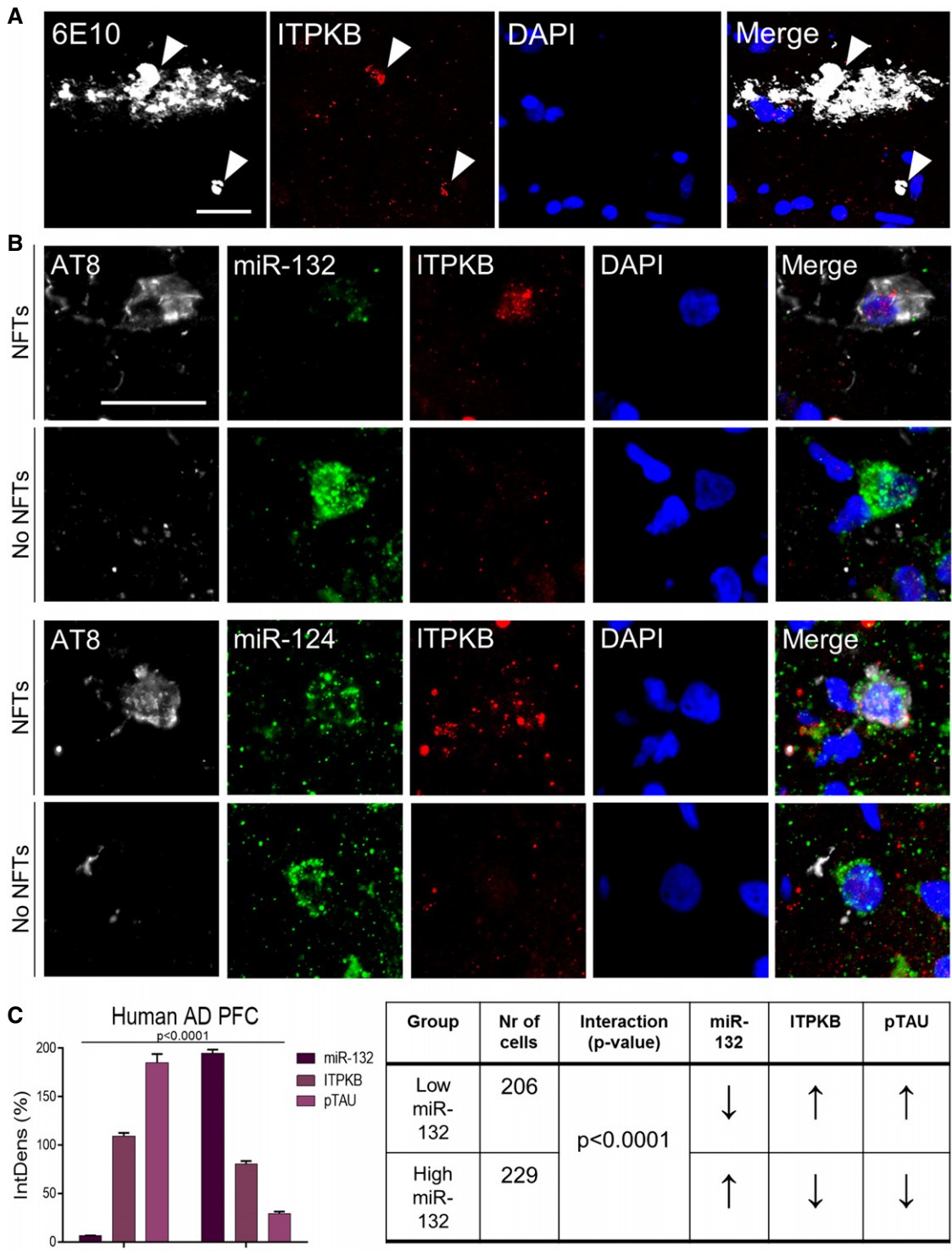

**Figure 7.  miR-132/ITPKB expression profile in human AD prefrontal cortex.**

A  Double immunofluorescence staining of amyloid plaques (6E10) and ITPKB in AD prefrontal cortex. Scale bar, 50 μm. Arrowheads indicate ITPKB immunopositivity.

B  miR-132 FISH coupled with double immunofluorescence against hyperphosphorylated TAU (AT8)-containing neurofibrillary tangles (NFTs) and ITPKB in AD prefrontal cortex. miR-124 was used as a control. Scale bar, 50 μm.

C  Quantification of miR-132, ITPKB and hyperphosphorylated TAU (pTAU) signal in single neurons. Integrated intensity values of each signal per cell were normalized to the mean integrated density of each signal across all the cells analyzed (see Materials and Methods). Sample size (AD patients), *n* = 3. Values are presented as mean ± SEM. Two-way ANOVA was used. Quantifications are summarized in the table provided. Arrow directions refer to the comparison of each normalized signal to the same signal in the other group (in the "low miR-132" group comparisons are made to the "high miR-132" group and vice versa).

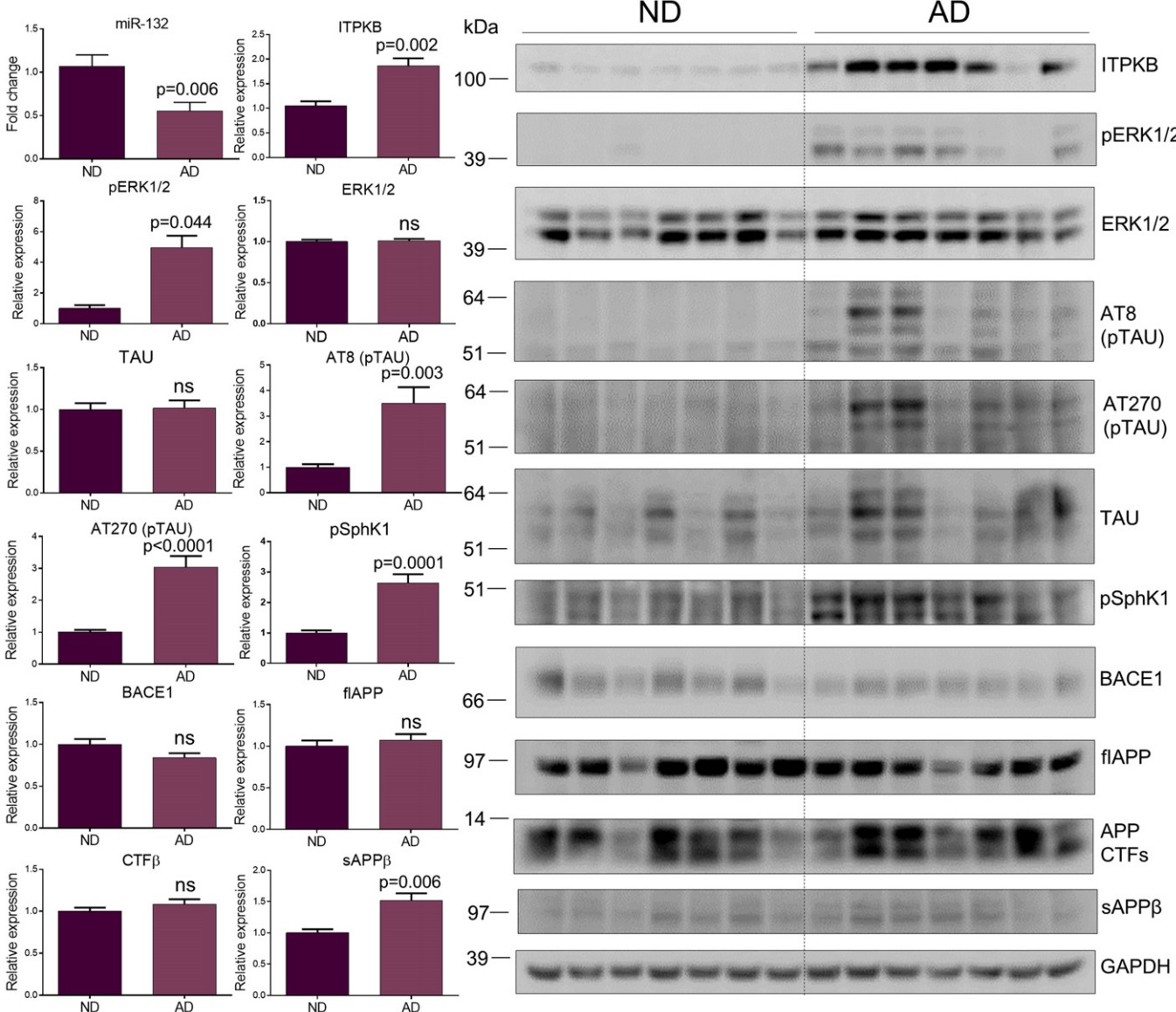

**Figure 8.   miR-132/ITPKB pathway assessment in human AD hippocampus.**
Semi-quantitative PCR of miR-132 and Western blot analysis of ITPKB, phosphorylated (pERK1/2) and total ERK1/2, phosphorylated (AT8, AT270) and total TAU, phosphorylated sphingosine kinase 1 (pSphK1), BACE1, full length APP (flAPP), APP CTFs and sAPPβ in human AD hippocampi (AD) compared to non-demented control samples (ND). Sample size, $n = 39$ for AD and $n = 15$ for ND. Values were normalized to ND and presented as mean ± SEM. Student's $t$-test was used.

Source data are available online for this figure.

mitigate the progressing neurodegenerative process in AD patients.

# Materials and Methods

### Animal procedures

All animal experiments were approved by the ethical committees of KU Leuven and UZ Leuven (LA1210596). Eight-week-old males were used in all stereotaxy protocols. Sample size was estimated in pilot studies prior to each experimental approach.

### Intracerebroventricular injections

The intracerebroventricular injections were performed as previously described (Jimenez-Mateos et al, 2011) using the following stereotactic coordinates: AP—0.1 mm, ML—1.0 mm, and DV—3.0 mm (from the skull). For miR-132 downregulation, mice were infused with 2–3 μl of miR-132 antagomiR (locked nucleic acid (LNA)-, 3′-cholesterol-modified oligonucleotide) (Exiqon, Denmark) in artificial cerebrospinal fluid (CSF) (Harvard Apparatus, USA). Control mice received either a scrambled LNA oligonucleotide or CSF. In total, four injections of 0.66 nmol each were performed per animal with 1-month intervals over a total period of 4 months. Analysis of

antagomiR-132-injected animals was performed at 6 months of age. For miR-132 overexpression, mice received either a miR-132 mimic or a negative control oligonucleotide (Dharmacon, GE Healthcare, Belgium) in a mix with lipofectamine 2000 (at a 1:1 volume ratio) (Thermo Fischer Scientific, Belgium). Injections of 150 pmol oligo each were performed once a week for 1 month in total. Analysis of miR-132 mimic-injected animals was carried out at 3 months of age. For *in vivo* ITPKB downregulation, 11-week-old mice received either an siRNA against ITPKB (Accell mouse SMARTpool ITPKB siRNA, Dharmacon, GE Healthcare, Belgium) or a control oligonucleotide (2 injections of 1 μg each with 4 days interval at a 1:1 volume ratio in lipofectamine) and sacrificed at 3 months of age. Randomization of injectates was employed for all injection sessions, and animals were randomly allocated to each treatment.

## RNA isolation, reverse transcription, and real-time PCR

For RNA analysis, hippocampi were processed using the miRVana Paris Kit (Life Technologies, Belgium) according to the manufacturer's instructions. Briefly, tissue was homogenized (or cells were lysed) in 300 μl cell disruption buffer supplemented with protease and phosphatase inhibitors. Following denaturation, addition of acid phenol:chloroform, incubation, and centrifugation, 1.25 volumes of ethanol 100% were added to the aqueous phase. The samples were then loaded on miRVana spin columns and processed according to the manufacturer's instructions. Reverse transcription of 200 ng (mRNA) or 100 ng (miRNA) RNA was performed using the Superscript II reverse transcriptase (Invitrogen, Life Technologies Europe, Belgium) for protein-coding transcripts and the Universal cDNA synthesis kit (Exiqon, Denmark) for miRNAs. Real-time semi-quantitative PCR was performed using the LightCycler 480 Sybr Green (Roche Diagnostics, Belgium) for coding transcripts and the Sybr Green mastermix and LNA PCR primers (Exiqon, Denmark) for miRNAs. The primer sequences can be found in Appendix Table S1. Primers. Cp (crossing points) were determined by using the second derivative method. Fold changes were calculated with the ΔΔCt method (Livak & Schmittgen, 2001).

## Western blotting

For protein analysis, tissue was homogenized (or cells were lysed) in 300 μl cell disruption buffer as described above. Following centrifugation, the supernatants were diluted in 1× SDS–PAGE loading buffer containing 5% β-mercaptoethanol, boiled for 5 min at 96°C and centrifuged briefly. About 15 μg (mouse), 15 μg (cells), or 30 μg (human) of protein were finally electrophoresed on NuPAGE 10% Bis-Tris gels (Invitrogen, Life Technologies, Belgium). Following electrotransfer, the nitrocellulose membranes were blocked in blocking solution (5% milk powder in TBS-Tween 0.1%) and then incubated with primary antibodies in blocking solution overnight. Blots were incubated with the appropriate secondary antibody in blocking solution for 1 h at room temperature and then developed using chemiluminescence (Perkin Elmer, USA).

## Fluorescent *in situ* hybridization and immunofluorescence

The *in situ* hybridization protocol was adapted from Silahtaroglu *et al* (2007); Papadopoulou *et al* (2015). Briefly, 20 μm (mouse)- or

18 μm (human)-thick brain sagittal cryosections were postfixed in ice-cold PFA 4% for 15 min, acetylated for 1 min, and prehybridized in 50% formamide, 5×SSC and 500 μg/ml yeast t-RNA (prehybridization buffer) for 1 h at 60°C. Hybridization was performed with 42 nM of miR probe or scrambled negative control (5′-fluorescein, LNA, 2′-OMe oligonucleotides) (Ribotask, Denmark) in prehybridization buffer for 1 h at 70°C. Following posthybridization washes in 0.2× SSC (70°C), 0.5× SSC (70°C), 2× SSC (room temperature) and incubation in 3% H$_2$O$_2$ for 7 min, sections were incubated in 0.5% BSA and 0.5% blocking reagent (Roche Diagnostics, Belgium) (blocking solution 1). Finally, sections were probed with an antifluorescein HRP-conjugated antibody (Roche Diagnostics, Belgium) in blocking solution 1 and signals were developed in TSA Plus Fluorescein reagent (Perkin Elmer, USA). For immunofluorescence, samples were subsequently boiled in 10 mM sodium citrate, 0.05% Tween-20, pH 6.0 for antigen retrieval, incubated in 2% normal goat serum in 0.5% TBS-Triton X-100 for 1 h (blocking solution 2), probed with the primary antibodies in blocking solution 2 at 4°C overnight and with the appropriate secondary antibodies for 2 h at room temperature. Finally, sections were incubated in DAPI (Sigma-Aldrich, Belgium) and mounted in Mowiol.

## Image acquisition and quantification

Images (z-stacks) were acquired using a Nikon A1R Eclipse Ti confocal microscope. The FIJI software was employed for image processing and quantification. In all quantifications, four sections per sample were used with 100 μm distance between each two sections and average values for each sample were calculated. For amyloid burden, a region of interest was manually drawn around each plaque and the total area occupied by plaques in hippocampus or cortex was calculated per sample. The mean of the total area in scramble-injected mice was set to 100, and all values were normalized to 100%. For miR-132 and miR-124 total signal quantification in human prefrontal cortex, integrated density of miR staining in grey matter was calculated. The mean of the integrated density in ND samples was set to 100, and all values were normalized to 100%. For single-cell colocalization analysis of miR-132, ITPKB, and pTAU, the mean integrated density for each of the three signals (miR-132, ITPKB, and AT8) across all the cells analyzed (435 cells) was set to 100. The integrated intensity of each of the three signals for each cell was then normalized against the appropriate mean to 100%. Eventually cells were grouped together into two separate groups based on the miR-132 normalized intensity (high or low). The cell populations analyzed spread equally across the three AD PFC samples. All image quantifications were performed in a single-blind manner.

## Antibodies

Western blotting: mouse anti-CDK5 (1:1,000, sc-6247, Santa Cruz, USA); rabbit anti-GSK3b (9315), rabbit anti-ERK1/2 (9102), rabbit anti-pERK (9101s), rabbit anti-BACE1 (1:1,000, s5606, Cell Signaling, USA); mouse TAU-5 (anti-total TAU) (1:1,000, 577801, Calbiochem, Merch Chemicals, Belgium); mouse AT8 (MN1020), AT100 (MN1060), AT180 (MN1040), AT270 (MN1050) (1:500, Pharmingen, BD Biosciences, Belgium); rabbit anti-ITPKB (1:500, 12816-1-AP, Proteintech, Germany); rabbit B63 (anti-flAPP, anti-CTFs) (De

Strooper *et al*, 1993); rabbit anti-sAPPβ wild-type (1:500, 813401, Covance BioLegend, USA); mouse anti-sAPPβ Swe (1:250, 10321, IBL, Germany); and anti-pSphK (1:500, SP1641, ECM Biosciences, USA).

Immunofluorescence: mouse AT8 (1:100, Cell Signaling, USA); rabbit anti-ITPKB (1:50, Proteintech, Germany); and mouse 6E10 (amyloid staining) (1:150, 803002, BioLegend, USA).

## Cell culture and transfections

For miR-132 downregulation, a miR-132 antisense hairpin inhibitor was used (GE Healthcare, Belgium) to transfect HEK293-APP$^{swe}$ using lipofectamine. Control transfections were performed using a negative hairpin oligonucleotide (GE Healthcare, Belgium). For ITPKB knockdown, a 27-mer siRNA duplex against human ITPKB or a scrambled negative control were used (Origene, USA) to transfect HEK293-APP$^{swe}$ using lipofectamine. Briefly, cells were seeded on 12-well plates at a cell density of 250,000 cells per well. Twenty-four hours after seeding, cells were transfected and finally harvested 72 h thereafter.

## Correlation analysis

To correlate the expression of hsa-miR-132-3p to the expression of its predicted targets in the human prefrontal cortex, the normalized miRNA deep sequencing data were obtained from Lau *et al* (2013) and the normalized mRNA microarray data from Bossers *et al* (2010), both of which were performed on the prefrontal cortex of the same patient cohort from the Netherlands Brain Bank (Amsterdam, Netherlands). The data for the six Braak 0-I and six Braak V-VI patients were derived from the full normalized mRNA microarray dataset. Pearson's correlation coefficient and accompanying *P*-values and Benjamini–Hochberg-corrected *P*-values were calculated for the expression of hsa-miR-132-3p versus all detected mRNAs. A total of 960 transcripts were mapped to the mRNA profiling dataset out of a total of 1332 initially predicted (Targetscan 7.0) miR-132 targets.

## Luciferase assay

The 3′UTRs were obtained by Gen9, Inc., USA. Subcloning into the psiCHECK plasmid, transfection into wild-type HEK293 cells, and luciferase assay were performed as previously described (Salta *et al*, 2014).

## Aβ extraction and ELISA

For Aβ isolation and quantification in hippocampi, the three-step Aβ extraction protocol was adapted from Shankar *et al* (2011). Briefly, TBS extracts were prepared by mechanical homogenization of tissue in TBS supplemented with protease and phosphatase inhibitors and subsequent centrifugation at 88,000 *g* in a TLA 100.4 rotor on a Beckman for 1 h at 4°C. TBS-Triton X-100 extracts were obtained by rehomogenizing the pellets from the previous step in TBS with 1% Triton X-100, followed by ultracentrifugation as before. Finally, rehomogenization of the previous pellet in 88% formic acid, sonication, overnight incubation at 4°C, and neutralization in non-buffered Tris 1 M yielded the formic acid extract. For Aβ quantification *in vitro*, HEK293-APP$^{swe}$ (kind gift from Christian Haass) were grown on poly-l-lysine plates in DMEM/F12 with 10% FBS for 48 h

following transfection. Medium was replaced with DMEM/F12 with 1% FBS, and cells (for protein and RNA isolation) and medium (for Aβ quantification) were collected 24 h thereafter. Aβ ELISA measurements were performed as previously described (Thathiah *et al*, 2013). Values were normalized to extract protein concentration (for TBS extracts) or cell protein concentration (for HEK293-APP$^{swe}$).

## Human samples

Hippocampal tissue samples were obtained from the London Neurodegenerative Diseases Brain Bank. These samples were pathologically confirmed but not further categorized according to Braak stage (Thathiah *et al*, 2013). Prefrontal cortex tissue blocks were obtained from the Banner Sun Health Research institute (Arizona, USA). All brain samples were collected according to legislation and ethical boards of the respective brain banks. The human study was evaluated and approved by the ethical committees of Leuven University and UZ Leuven (Thathiah *et al*, 2013). Protein extracts were prepared in 2% SDS supplemented with protease and phosphatase inhibitors.

**Expanded View** for this article is available online.

## Acknowledgements

Evgenia Salta is an FWO postdoctoral fellow. Work in the De Strooper laboratory is supported by European Research Council (ERC) grant ERC-2010-AG_268675, the Fonds voor Wetenschappelijk Onderzoek (FWO), KU Leuven, VIB, and a Methusalem grant from KU Leuven and the Flemish Government. Bart De Strooper is the Bax-Vanluffelen Chair for Alzheimer's Disease and is

**The paper explained**

**Problem**

Several microRNAs are deregulated in AD brain, suggesting their functional interrelationship to AD pathology. More specifically, microRNA-132 (miR-132), a potent neuroprotective molecule, is robustly down-regulated early on in AD patients. We address here the functional implications of miR-132 loss in AD brain and investigate the potential of using miR-132 in the clinic in the future.

**Results**

We show that miR-132 is a potent early regulator of both Aβ and TAU pathology via its target ITPKB, which further activates ERK1/2 leading to increased BACE1 activity and TAU phosphorylation. Lowering miR-132 in the brain of an AD mouse model aggravates both arms of the pathology, while boosting miR-132 levels ameliorates both amyloidosis and TAU hyperphosphorylation. Similar changes are observed in human AD brain, where we observed decreasing miR-132, increasing ITPKB and phospho-ERK1/2, enhanced BACE1 processing of APP, and TAU hyperphosphorylation. The data provide functional evidence for the importance of the previously reported downregulation of miR-132 in AD brain.

**Impact**

Identifying novel mechanisms that contribute to the preclinical phases of AD pathology is of critical importance for early intervention. Restoring miR-132 levels in human diseased brain might represent a novel therapeutic target in AD.

supported directly by the Opening the Future campaign of the Leuven Universiteit Fonds (LUF). We are grateful to Veronique Hendrickx and Jonas Verwaeren for animal husbandry, Ashley Lu and Mark Fiers for assistance with bioinformatics analysis, Tobias Engel, Eva Jimenez-Matteos, David Henshall, and Tom Jaspers for help with stereotaxy, An Snellinx, Katrien Horré, and Katleen Craessaerts for technical support, and Carlo Sala Frigerio and Lujia Zhou for feedback. APPPS1 mice were a kind gift from Mathias Jucker, DZNE, Germany. Confocal microscope equipment was acquired through a Hercules Type 1 AKUL/09/037 to Wim Annaert.

## Author contributions

ES, AS, and BDS conceived the study and performed the data analysis. ES and BDS wrote the manuscript. ES and EVE performed the experiments. AS and EVE performed the ICV injections. All co-authors provided discussion and data interpretation and contributed to the final version of the manuscript.

## Conflict of interest

The authors declare that they have no conflict of interest with the context of this study. BDS is consultant for Janssen pharmaceutica and Remynd. He receives research funding from Janssen pharmaceutica, but not for the work presented in the current manuscript.

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
