## [Review Process File · EMBO Molecular Medicine]

miR-132 loss de-represses ITPKB and aggravates amyloid and TAU pathology in Alzheimerís brain

Evgenia Salta, Annerieke Sierksma, Elke Vanden Eynden, Bart De Strooper

Corresponding author: Bart De Strooper, Katholieke Universiteit Leuven

Review timeline:	Submission date:	19 April 2016
	Editorial Decision:	26 May 2016
	Revision received:	01 June 2016
	Editorial Decision:	28 June 2016
	Revision received:	06 July 2016
	Accepted:	06 July 2016

Transaction Report:

Editor: Céline Carret

1st Editorial Decision

26 May 2016

Thank you for the submission of your manuscript to EMBO Molecular Medicine. We have now heard back from the two referees whom we asked to evaluate your manuscript.

As you will see from the comments below, both referees find the data well done and convincing. However, both have a number of suggestions to improve the study; of great interest to us is the suggestion to provide cognitive test behaviour analyses, which we really hope, you will be able to perform.

We would welcome the submission of a revised version for further consideration and depending on the nature of the revisions, this may be sent back to the referees for another round of review. Please be reminded that EMBO only allows one round of major revision. Revised manuscripts should be submitted within three months of a request for revision; they will otherwise be treated as new submissions, except under exceptional circumstances in which a short extension is obtained from the editor.

I look forward to seeing a revised form of your manuscript as soon as possible.

***** Reviewer's comments *****

Referee #1 (Remarks):

The manuscript by Salta and colleagues shows that acute miR132 downregulation results in increased pTau and Abeta pathologies, and that this occurs through the miR132 target ITPKB. The data in mice is mimicked in the human AD samples, and opens the door for miR132 as a therapy for AD as it targets both arms of the disease pathway. The mice studies, the in vitro work validating the target, and most of the human patient data is convincing.

For the histological co-localization of miR132 elevation with decreased ITPKB, and miR132 loss with ITPKB, its difficult to know how frequent this association is from the photomicrographs provided. It would be helpful to add n's to the figure (x out of y cells examined).

Is there an effect of the miR132 KD or over-expression on the cognitive measures in this model? Did they appear earlier as could be expected from the exacerbation of the pTau and amyloid deposition?

Minor comment

- some of the references are lacking the full citation (for example, the journal)

Referee #2 (Remarks):

The authors show data to support a disease cascade of miR-132 downregulation to upregulation of ITPBK leading to Tau/Abeta pathology via enhanced Erk activity. The Tau phenotype confirms some of the findings by Smith et al (HMG 2015) although the new data argue for a distinct mechanism. The findings are interesting and overall convincing. However, some of the data is based on reanalyzing previous data sets (Lau et al, 2013) or confirmatory (e.g. Stygelbout et al, 2014). The most exciting aspect is amelioration of Abeta pathology upon miR-132 injection. It would be nice to extend this aspect, e.g. does this treatment improve cognition in the mouse model?

Major points:

Fig 5: The effects on ITPKB are modest, which is in line with typical miRNA effects. Are similar changes by knockdown or overexpression of ITPKB sufficient to cause the observed effects on Abeta? In Fig 5B the authors should also show ITPB protein levels in this system: is the "rescue" accompanied by normal or dramatically reduced ITPKB levels??

Fig 8: In the text it says that phospho-Tau is elevated, while total Tau remains the same, which is expected from previous reports. However, the figure shows exactly the opposite result. How can that be explained?

Is there a correlation of ITPKB with Abeta/tau load in AD brains?

Minor points:

Legend 4B mentions a "semi-quantitative PCR". From the methods I would call it quantitative unless the results were obtained from EtBr gel stains.

Fig 5: The dotted lines between groups should be removed unless the blots were cut. Resort panels to fit to text (5B).

Response to the reviewers

We would like to thank the referees for their critical and constructive reports. We have now incorporated additional experiments and analyses to solidify our observations and strengthen our conclusions. For a detailed response please see below.

Referee #1 (Remarks):

The manuscript by Salta and colleagues shows that acute miR132 downregulation results in increased pTau and Abeta pathologies, and that this occurs through the miR132 target ITPKB. The data in mice is mimicked in the human AD samples, and opens the door for miR132 as a therapy for AD as it targets both arms of the disease pathway. The mice studies, the in vitro work validating the target, and most of the human patient data is convincing.

We appreciate the reviewer's interest in our work and his/her effort to contribute to its improvement. We address the comments in our point-by-point response below:

For the histological co-localization of miR132 elevation with decreased ITPKB, and miR132 loss with ITPKB, its difficult to know how frequent this association is from the photomicrographs provided. It would be helpful to add n's to the figure (x out of y cells examined).

Following the reviewer's suggestion, we now included a detailed quantification of the pertinent stainings (please refer to revised Figure 7), where we divide the cells analyzed into two distinct groups (based on low or high miR-132 signal) and also report the accompanying signals for ITPKB and hyperphosphorylated TAU per group. In the additional table (Figure 7), we provide all the required information on the total numbers of the analyzed cells and the statistical significance of the differences among groups. Also, a relevant section has been added to the Materials and Methods explaining the methodology employed (please see "Image acquisition and quantification"). As already mentioned in the text, in cells with low miR-132 levels, ITPKB and phospho-TAU signals are high, whereas this expression profile is reversed in neurons with higher miR-132 expression.

Is there an effect of the miR132 KD or over-expression on the cognitive measures in this model? Did they appear earlier as could be expected from the exacerbation of the pTau and amyloid deposition?

We agree that delineating the functional effect of miR-132 KD in our AD mouse model is important when further exploring the potential of miR-132 as a therapeutic approach for AD. We would like to point out that other work has provided already initial indications on the behavioral effects of miR-132 modulation in AD (Smith et al, 2015, miR-132/212 deficiency impairs tau metabolism and promotes pathological aggregation in vivo, Hum Mol Genet), where miR-132 overexpression in 3xAD mice significantly improved long-term memory. Additionally, the same group has also previously shown that miR-132 KO impairs memory (Hernandez-Rapp et al, 2015, Memory formation and retention are affected in adult miR-132/212 knockout mice, Behavioural Brain Research). Hence, it is not unreasonable to assume that the miR-132 KD in our mice would exacerbate the behavioral deficits in these animals. However, our ICV miR-132 down regulation paradigm takes four months, which together with the assembly of cohorts big enough for behavioral testing and analysis, would require at least another year to finalize. We hope that the reviewer will agree that our conclusions on the regulatory cascade starting from miR-132 and ITPKB and eventually leading to A β and pTAU have now been substantially strengthened in the revised manuscript and that carrying out a time-consuming behavioral study would be beyond the scope of the current work. We ascertain that we have planned a further follow-up project to address this concern in the frame of a large study using multiple mouse models and different routes and schemes of administration to explore the therapeutic potential of miR-132 in the context of different AD models.

Minor comment:

- some of the references are lacking the full citation (for example, the journal)

The references with missing information have been corrected accordingly.

Referee #2 (Remarks):

The authors show data to support a disease cascade of miR-132 downregulation to upregulation of ITPKB leading to Tau/Abeta pathology via enhanced Erk activity. The Tau phenotype confirms some of the findings by Smith et al (HMG 2015) although the new data argue for a distinct mechanism. The findings are interesting and overall convincing. However, some of the data is based on reanalyzing previous data sets (Lau et al, 2013) or confirmatory (e.g. Stygelbout et al, 2014). The most exciting aspect is amelioration of Abeta pathology upon miR-132 injection.

We appreciate the reviewer's critical and at the same time constructive view of our manuscript. We now include new experiments to cement our novel finding that miR-132 exerts an ITPKB-mediated effect on A β *in vivo* (please refer to the revised Figure 5). All the reviewer's comments are addressed in our point-by-point response below.

It would be nice to extend this aspect, e.g. does this treatment improve cognition in the mouse model?

We definitely agree with the reviewer on the significance of the behavioral impact of miR-132 overexpression in the AD mice. Interestingly, initial indications on the behavioral effects of miR-132 modulation in AD have been provided by Smith et al (Smith et al, 2015, miR-132/212 deficiency impairs tau metabolism and promotes pathological aggregation *in vivo*, Hum Mol Genet), where miR-132 overexpression in 3xAD mice significantly improved long-term memory. Moreover, the same group has also previously shown that miR-132 KO impairs memory (Hernandez-Rapp et al, 2015, Memory formation and retention are affected in adult miR-132/212 knockout mice, Behavioural Brain Research). Hence, we believe that ICV delivery of miR-132 mimics may potentially ameliorate the behavioral deficits normally observed in our AD mice. However, there are certain limitations to performing these experiments in the context of the current revision. Please note that Smith et al carried out their behavioral testing at 12 months of age, when cognitive deficits are well established in the 3xAD mice, similarly to our mouse model, where most of the cognitive and memory readouts start to be impaired only from 8-9 months onwards. Consequently, if properly performed, such a behavioral study would -realistically speaking- require at least one year (including mobilizing a sufficient large colony, aging of mice, ICV injections, behavioral testing and data analysis). We hope that the reviewer will appreciate the additional experiments/analyses added in the revised manuscript and will agree that the mechanism we put forward here is interesting and highly relevant for AD. We ascertain that we have planned a further follow up project to address this concern in the frame of a large study using multiple mouse models and different routes and schemes of administration to explore the therapeutic potential of miR-132 in the context of different AD models.

Major points:

Fig 5: The effects on ITPKB are modest, which is in line with typical miRNA effects. Are similar changes by knockdown or overexpression of ITPKB sufficient to cause the observed effects on Abeta?

In our *in vivo* experiments, ICV injections of miR-132 mimic oligonucleotides resulted in an ITPKB down regulation of about 50% and a concomitant decrease of insoluble A β of approximately 50-70%. Taking the reviewer's comment into consideration we have now performed additional *in vivo* experiments trying to mimic this observation by directly down regulating ITPKB levels in the APPPS1 hippocampus. As you can now appreciate in the revised Figure 5, down regulating ITPKB *in vivo* to an extent similar to the one achieved by miR-132 overexpression (50%) (Figure 5C) resulted in a similar decrease of insoluble A β (60-70%) (Figure 5D) clearly indicating that modulating ITPKB expression alone (within the range that would be considered physiological in relation to its regulation by miR-132) is indeed sufficient to elicit an effect on A β that phenocopies the one mediated by miR-132.

In Fig 5B the authors should also show ITPKB protein levels in this system: is the "rescue" accompanied by normal or dramatically reduced ITPKB levels??

We agree with the reviewer, that monitoring the actual levels of ITPKB upon any manipulation is of crucial importance for the interpretation of our findings. Hence, we have included western blot analysis of ITPKB levels in our *in vitro* miR-132/ITPKB co-down regulation experiment (in the same samples used for the ELISA shown in Figure 5B) in the revised version of our manuscript (please refer to Figure EV3B). As shown in the corresponding quantification, the double transfection led to a normalization of ITPKB levels back to the control ones (non significant difference from control samples).

Fig 8: In the text it says that phospho-Tau is elevated, while total Tau remains the same, which is expected from previous reports. However, the figure shows exactly the opposite result. How can that be explained?

Our western blot analysis of human AD hippocampal samples indeed confirmed an increase in hyperphosphorylated TAU levels (as assessed by immunostaining with the antibodies AT8 and AT270). This is clearly indicated by the respective blot images in Figure 8 and quantified in the pertinent graphs on the left where all 39 AD and 15 control samples were taken into account. Concerning total TAU levels, although we do agree with the reviewer that some AD samples do show a trend for increase, final quantification and statistical analysis of TAU expression in all 54 samples did not confirm such a change. To our knowledge, TAU protein or mRNA levels do not increase in the AD brain (contrary to the TAU protein levels in CSF), as opposed to the levels of hyperphosphorylated TAU as already suggested by the groups of Trojanowski, Lee and Blennow (Bramblett et al, 1992, Lab Invest; Sjögren et al, 2001, J Neurol Neurosurg Psychiatry), which puts forward TAU hyperphosphorylation rather than increased TAU synthesis as one of the mechanisms of pathogenesis in AD. Despite the fact that the overall TAU levels remain unchanged across the groups in our study as well, we cannot exclude that there might always be a small subpopulation of AD patients in which TAU protein is increased (or decreased).

Is there a correlation of ITPKB with Abeta/tau load in AD brains?

As already mentioned in the text, increased ITPKB levels have been previously reported in three independent studies (Bossers et al, 2010; Saetre et al, 2011; Stygelbout et al, 2014), while a spatial association between ITPKB and amyloid plaques has also been demonstrated (Stygelbout et al, 2014). Here, we have confirmed both the increase in ITPKB protein levels in our human AD hippocampal cohort and the association between ITPKB and amyloid and tangle (we have now included quantifications, please refer to revised Figure 7C) staining in human AD prefrontal cortex. Using the protein expression values of ITPKB and pTAU (AT270) we now carried out a correlation analysis to assess whether the association we observed in our western blots reflects an actual correlation between the two or not. Indeed, we found that ITPKB and hyperphosphorylated TAU are significantly correlated in human AD and control hippocampi (Pearson's correlation coefficient: 0.7; $p=0.002$) as opposed to ITPKB versus total TAU (Pearson's correlation coefficient: -0.2; $p=0.44$). This information has now been included in the text (please refer to "The miR-132/ITPKB pathway is disrupted in human AD brain" section). We do not have quantitative A β levels in the brain of these patients to generate a similar correlation with regard to A β but the strong negative effect of the *in vivo* ITPKB down regulation on A β levels in the APPPS1 hippocampus serves now as an additional indication of the functional relevance of this process (revised Figure 5C, D).

Minor points:

Legend 4B mentions a "semi-quantitative PCR". From the methods I would call it quantitative unless the results were obtained from EtBr gel stains.

We do not measure absolute cDNA levels in our real-time PCR, but normalize to the scramble-injected control group and express the final values as fold changes compared to the levels in the controls which has been set to 1. Therefore, we would like for consistency reasons to keep referring to the method employed here as "semi-quantitative".

Fig 5: The dotted lines between groups should be removed unless the blots were cut. Resort panels to fit to text (5B).

The dotted lines between different groups in western blots have been removed (except for Figure 8, where the length of the blot panel would otherwise make it too difficult for the reader to distinguish ND/AD in all the blots) and the legends of the x axis in Figure 5B have been corrected according to the reviewer's suggestion.

2nd Editorial Decision

28 June 2016

Thank you for the submission of your revised manuscript to EMBO Molecular Medicine. We have now received words from the referees that were asked to re-assess it. As you will see the reviewers are now globally supportive and I am pleased to inform you that we will be able to accept your manuscript pending editorial final amendments:

Please submit your revised manuscript within two weeks.

I look forward to reading a new revised version of your manuscript as soon as possible.

***** Reviewer's comments *****

Referee #1 (Comments on Novelty/Model System):

I have no additional comments.
I am satisfied with the revision.

Referee #2:

Is suitable for publication.

Corresponding Author Name: Bart De Strooper

Manuscript Number: EMM-2016-06520